# Oviposition Preference and Age-Stage, Two-Sex Life Table Analysis of *Spodoptera frugiperda* (Lepidoptera: Noctuidae) on Different Maize Varieties

**DOI:** 10.3390/insects14050413

**Published:** 2023-04-26

**Authors:** Qiang-Yan Zhang, Yan-Lei Zhang, Peter Quandahor, Yu-Ping Gou, Chun-Chun Li, Ke-Xin Zhang, Chang-Zhong Liu

**Affiliations:** 1Biocontrol Engineering Laboratory of Crop Diseases and Pests of Gansu Province, College of Plant Protection, Gansu Agricultural University, Lanzhou 730070, China; zhangqiangyan2@163.com (Q.-Y.Z.); pgy16868@163.com (Y.-L.Z.); gouyp@gsau.edu.cn (Y.-P.G.); lcc@gsau.edu.cn (C.-C.L.); zhangkx199404@163.com (K.-X.Z.); 2CSIR—Savanna Agricultural Research Institute, Tamale P.O. Box 52, Ghana; quandooh@yahoo.com

**Keywords:** *Spodoptera frugiperda*, special maize, common maize, host preference, life cycle

## Abstract

**Simple Summary:**

The fall armyworm (FAW), *Spodoptera frugiperda* (JE Smith), is an invasive pest that feeds on a wide range of host plants, especially maize, threatening agricultural production and food security in China. Thus, understanding its host plant preferences is of great significance for its population prediction and management. In this study, the oviposition preferences of *S. frugiperda* on ten special maize and ten common maize varieties were tested, and the larval development, adult reproduction, and population parameters of *S. frugiperda* were also examined. Generally, *S. frugiperda* oviposited and completed its life cycle across all maize cultivars. However, *S. frugiperda* females preferred the special maize varieties over common maize varieties for oviposition. Moreover, faster growth, higher pupal weight, *S. frugiperda* adult fecundity, net reproductive rate (*R*_0_), intrinsic rate of increase (*r*), and finite rate of increase (*λ*) occurred on the special maize varieties. However, the lowest oviposition preference, *R*_0_, *r*, *λ*, and longest *T* occurred on Zhengdan 958, suggesting that Zhengdan 958 is a less suitable host compared to the other tested maize varieties. Our findings can provide a reference for the detection of population dynamics and rational planting of maize and help to refine laboratory rearing protocols.

**Abstract:**

Host plants play an important role in the growth, development, and reproduction of insects. However, only a few studies have reported the effects of maize varieties on the growth and reproduction of *S. frugiperda*. In this study, a free-choice test was used to evaluate the oviposition preferences of female adults on ten common maize varieties and ten special maize varieties. The population fitness of *S. frugiperda* on six different maize varieties was also examined using the age-stage, two-sex life table method. The results showed that *S. frugiperda* oviposited and completed its life cycle across all maize cultivars. Moreover, the *S. frugiperda* females exhibited a significantly higher oviposition preference on the special maize varieties than on the common maize varieties. The highest number of eggs and egg masses occurred on Baitiannuo and the lowest on Zhengdan 958. The egg + larval stage, preadult, pupal stage, adult, APOP, TPOP, and total longevity of *S. frugiperda* were significantly shorter on the special maize varieties than on the common maize varieties. The fecundity, oviposition days, pupal weight, and hatching rate of *S. frugiperda* were significantly higher on the special maize varieties than on the common maize varieties. Specifically, *S. frugiperda* had the highest fecundity, female, and male pupal weight on Baitiannuo. Moreover, the net reproductive rate (*R*_0_), intrinsic rate of increase (*r*), and finite rate of increase (*λ*) of *S. frugiperda* were the greatest on Baitiannuo, whereas the shortest mean generation time (*T*) occurred on Zaocuiwang. The lowest *R*_0_, *r*, and *λ*, and longest *T* occurred on Zhengdan 958, suggesting that Zhengdan 958 is a non-preferred host plant compared to the other tested maize varieties. The findings of this study can provide a reference for the rational planting of maize and provide basic scientific information for the management of *S. frugiperda*.

## 1. Introduction

The fall armyworm (FAW), *Spodoptera frugiperda* (J.E. Smith) (Lepidoptera: Noctuidae), is a tropical and subtropical America native invasive pest [1,2]. It is a major pest worldwide, which causes economic damage to the leaves and stems of more than 350 different host plants such as *Poaceae* (106 species), *Asteraceae* (31 species), and *Fabaceae* (31 species) [3]. In a favorable environment, the FAW can increase fecundity, spread quickly, and seriously threaten crop production [4,5,6]. It has become a major threat to food security in developing countries. The United Nations’ efforts to improve the well-being and good health of everyone are highly endangered due to the worldwide invasion of the fall armyworm [7,8]. FAW invasion is a major concern in China. This is due to the increase in climate change, drought, and other uncontrolled climatic conditions, creating an ideal environment for the pests to grow and develop [8,9].

Typically, the FAW contains two biotypes: corn strain and rice strain [10]. The host plant’s variation may significantly influence the growth, developmental time, survival rate, and reproduction rate of herbivorous insects [11]. Different maize varieties can have an effect on the oviposition, feeding and preference characteristics, development, and reproduction parameters of *S. frugiperda* [12,13,14,15,16]. Globally, maize (*Zea mays* L.) production is increasing at a faster rate in terms of tons per year. Maize is in high demand due to its significant contribution to food security and income generation for Chinese farmers [17]. China is the world’s second-largest producer of maize [18]. In 2019, the planting area of maize nationwide reached 41 million ha [16]. Unfortunately, by October 2019, *S. frugiperda* had spread to more than 20 Chinese provinces (autonomous regions, municipalities), causing damage of more than one million hectares to local maize and accounting for 98.6% of the total area where *S. frugiperda* occurred [9]. *S. frugiperda* larva causes extensive defoliation and interferes with the normal growth of maize plants by feeding on young maize leaf whorls, tassel, and ears [19,20,21].

A life table is one of the important tools for evaluating pests’ relative suitability of host plant [22]. However, the traditional life table, which ignores the male population, different developmental stages, and individual differences, is reported to influence the description of insects’ population characteristics [23]. The age-stage, two-sex life table, which eliminates several of these inherent errors, is greatly recommended, as it contains data from both sexes in the population [24].

This study was based on the hypothesis that different maize varieties will affect the oviposition selectivity and development of *S. frugiperda*. Therefore, the effects of different maize varieties on the oviposition selectivity and biological characteristics of *S. frugiperda* were determined by the free-choice test and age-stage, two-sex life table method. Our findings can provide a reference for the rational planting of maize and provide basic information for the management and control of *S. frugiperda*.

## 2. Materials and Methods

### 2.1. Insect Population Rearing

*S. frugiperda* larvae were originally obtained from a maize field at Jingyuan county (36°34′16″ N, 104°40′36″ E), Gansu Province, China, in September 2019. These larvae were reared on artificial diet in insect cages as described by Wang et al. (2019) [25]. Insects were maintained in a controlled environment at 25 ± 1 °C under 75 ± 5% relative humidity (RH) and a photoperiod of 16:8 h (L:D). *S. frugiperda* was continuously reared for ten generations for experiments.

### 2.2. Plant Materials and Growth Conditions

The study involved 20 maize varieties. There were 10 common maize varieties (Zhengdan 958, Wuke No. 2, Wuke No. 3, Wuke No. 4, Wuke 113, Wuke 187, Wuke 609, Honghe 26, Qingchu 107, Zhenghong No. 6) and 10 special maize varieties which included 4 sweet maize varieties (Shuangsecuibao, Zaocuiwang, Jingxiangyu, Jingtian No. 3), 3 sweet and waxy maize varieties (Heitiannuo, Tiancainuo, Baitiannuo), and 3 waxy maize varieties (Heinuo, Bainuo, Ziyunuo). These common and special maize seeds were provided by Jiuquan Jinhui Agricultural Development Co., Ltd. (Jiuquan, China).

The seeds were sown in pots (10 cm diameter) for oviposition selectivity evaluation. Two seeds were planted in each pot to ensure consistency between the egg area laid and the probability of selection. The oviposition selectivity was tested during the six leaf stages of maize. Healthy and pesticide-free host plants were used for experiments. The tested hosts for measuring the population growth and development were planted in the experimental field of Gansu Agricultural University (36°5′20″ N, 103°41′54″ E).

### 2.3. Oviposition Selectivity of S. frugiperda to Different Maize Varieties

The experiment involved three groups. Group 1: the oviposition selectivity of *S. frugiperda* on 10 common maize varieties; Group 2: the oviposition selectivity of *S. frugiperda* on 10 special maize varieties; Group 3: *S. frugiperda* oviposition selectivity on maize selected from Group 1 and 2 (Zhengdan 958, Wuke No.4, Wuke 113, Zaocuiwang, Baitiannuo, and Ziyunuo). The insect cage used was 40 × 40 × 40 cm^3^, protected by mesh. The experimental plants were kept, accordingly, in the insect cage. Host plants with similar height and leaf area were selected. Three days after emergence, ten pairs of adults (six pairs in the third group) were placed in the same cage. All adults were fed daily with a 10% (*v*/*v*) honey solution and replicated six times per group. Three days after laying eggs, the number of eggs and egg masses in each cage on different maize varieties were checked and recorded, and the percentage of eggs and egg masses were calculated as listed below:Percentage of eggs = (amount of egg laid on each plant/amount of egg laid on all plants) × 100%
Percentage of egg masses = (number of egg masses per plant/number of egg masses on all plants) × 100%

### 2.4. Effects of Maize Varieties on the Development and Population Parameters of S. frugiperda

Six maize varieties were selected for early oviposition selectivity based on high, medium, and low oviposition preference of *S. frugiperda* to common and special maize varieties. The common maize varieties involved Zhengdan 958, Wuke No. 4, and Wuke 113 whereas the special maize varieties involved Zaocuiwang, Baitiannuo, and Ziyunuo. Each treatment was observed and recorded from the egg stage (eggs were produced by the same female on the same day). Each experiment was repeated three times. After hatching, the larvae were gently collected individually in 10 mL plastic centrifuge tubes and fresh leaves from the six different tested maize varieties were provided until pupation. Until death, larval development and survival were recorded daily. On the second day of pupation, the pupae were weighed on an electronic balance (Lanzhou qianmei scientific analysis instrument Co., Ltd., Lanzhou, China), and then the sex was divided by male and female. They were then transferred individually to disposable plastic petri dishes (9.0 cm in diameter) containing humidity vermiculite until pupa emergence. Then, pupa duration was also recorded. On the first day of eclosion, healthy adults (♀:♂ = 1:1) were randomly selected in each group and moved separately into a 1000 mL cylindrical disposable plastic box (upper and lower mouth diameters of 14 and 11 cm, respectively, and with a height of 9 cm) and fed daily with 10% honey water absorbent cotton covered with wet gauze. The egg masses were collected and fresh leaves were provided every day. The cumulative number of eggs laid by each female daily, the duration of preoviposition, oviposition, and postoviposition, and the egg hatching rate were recorded until both female and male adults died.

### 2.5. Analysis of the Age-Stage, Two-Sex Life Table

The data of individual life table were analyzed by the TWO-SEX-MSChart program, which is based on age-stage, two-sex life table theory [26,27,28]. Fecundity *f_xj_* represents hatched eggs number produced by female adult at age *x*. The parameters of the age-stage two-sex life tables were calculated according to the following equations [29]:

The age-stage-specific survival rate (*S_xj_*): the probability that a newly laid egg will survive to age *x* and stage *j*:$$S_{xj}=\frac{n_{xj}}{n_{01}}$$

The age-specific survival rate (*l_x_*) represents the probability that a newborn egg will survive to age *x*:$$l_{x}=\overset{m}{\underset{j=1}{\sum }}S_{xj}$$
where *m* is the number of stages.

The age-specific fecundity of the population (*m_x_*): the number of eggs per individual at age *x*:$$m_{x}=\frac{\overset{m}{\underset{j=1}{\sum }}S_{xj}f_{xj}}{\overset{m}{\underset{j=1}{\sum }}S_{xj}}$$

The net reproductive rate (*R*_0_): the total offspring number that an individual can produce during its lifetime:$$R_{0}=\overset{\infty }{\underset{x=0}{\sum }}l_{x}m_{x}$$

The intrinsic rate of increase (*r*):$$\overset{\infty }{\underset{x=0}{\sum }}e^{-r\left(x+1\right)}l_{x}m_{x}=1$$

The finite rate of increase (*λ*):$$\lambda =e^{r}$$

The mean generation time (*T*):$$T=\frac{lnR_{0}}{r}$$

Age-stage-specific life expectancy (*e_xj_*) represents that an individual of age *x* and stage *j* is expected to live [30]:$$e_{xj}=\overset{\infty }{\underset{i=x}{\sum }}\overset{m}{\underset{y=j}{\sum }}S_{iy}^{\prime }$$

The $S_{iy}^{\prime }$ is the probability that an individual of age *x* and stage *j* will survive to age *i* and stage *y*.

Age-stage-specific reproductive value (*v_xj_*) represents the contribution of an individual of age *x* and stage *j* to the future population [31]:$$v_{xj}=\frac{e^{r\left(x+1\right)}}{S_{xj}}\overset{\infty }{\underset{i=x}{\sum }}e^{-r\left(i+1\right)}\overset{m}{\underset{y=j}{\sum }}S_{iy}^{\prime }f_{iy}$$

### 2.6. Statistical Analysis

Oviposition preference data were arranged and analyzed using Excel 2019 and IBM SPSS Statistics version 23.0 (Chicago, IL, USA). One-way analysis of variance (ANOVA) was used to examine the effects of different maize varieties on oviposition preference of female adults. Treatment means of oviposition preference were separated by Duncan’s new multiple range test. TWO-SEX-MSChart program was used to analyze the life table parameters of *S. frugiperda* feeding on different maize using the bootstrap method. Significant differences among the six maize varieties was estimated by the paired bootstrap test. The standard errors were determined by bootstrapping with 100,000 repetitions, with *p* < 0.05 considered statistically significant. The curves related to oviposition selectivity and pupal weight were drawn using OriginPro 2021 software (OriginLab Corporation, Northampton, MA, USA) and these curves (*S_xj_*, *l_x_*, *f_xj_*, *m_x_*, *l_x_m_x_*, *e_xj_*, and *v_xj_*) were generated by SigmaPlot 14.0 (Systat Software Inc., San Jose, CA, USA).

## 3. Results

### 3.1. Oviposition Selectivity of S. frugiperda on Different Maize Varieties

#### 3.1.1. Oviposition Selectivity of *S. frugiperda* on Common Maize Varieties

Female moths successfully laid eggs across all common maize varieties (Figure 1A). The results showed significant differences in the total number of eggs, egg masses per cage and percentage of eggs, and egg masses (Figure 1A,B). The highest number of eggs (904.50 ± 3.24 per cage), egg masses (7.50 ± 0.43 per cage), percentage of eggs (18.17%), and percentage of egg masses (17.15%) occurred on Wuke No.4. The number of eggs and percentage of eggs were significantly higher than those on other maize varieties (*p* < 0.05), followed by Honghe 26, which was significantly higher than those on other maize varieties (*p* < 0.05). However, female moths laid the lowest number of eggs (281 ± 2.44 per cage) and egg mass (2.83 ± 0.31 per cage) on Zhengdan 958, and the percentage of eggs and egg masses were 5.65% and 6.41%, respectively, which was significantly lower than those on other maize varieties (*p* < 0.05). The tendency of the total number of eggs and egg masses per cage was similar to the percentage of eggs and egg masses. There was an apparent oviposition preference on the common maize varieties. The results showed that *S. frugiperda* female adults preferred Wuke No.4 for oviposition but performed poorly on Zhengdan 958 (Figure 1).

#### 3.1.2. Oviposition Selectivity of *S. frugiperda* on Special Maize Varieties

The oviposition selectivity of *S. frugiperda* on the special maize varieties is shown in Figure 2. There were significant differences in the oviposition selectivity of females among the maize varieties. A significantly higher total number of eggs (1130.67 ± 4.55 per cage), egg masses (13.50 ± 0.50 per cage) (Figure 2A), percentage of eggs (24.33%), and egg masses (28.51%) (Figure 2B) were apparent in female adults reared on Baitiannuo (sweet–waxy maize) (*p* < 0.05), followed by Heitiannuo. In contrast, the females laid the lowest total number of eggs (163.00 ± 2.65 per cage), egg masses (2.17 ± 0.31 per cage), percentage of eggs (3.51%), and egg masses (4.55%) on Ziyunuo, which was significantly lower than those on others (*p* < 0.05). On the special maize varieties, the total number of eggs and egg masses per cage, percentage of eggs, and egg masses of *S. frugiperda* were higher on the sweet–waxy, followed by the sweet maize and the lowest occurred on the waxy maize. The difference between the three varieties was significant (*p* < 0.05). On the sweet–waxy maize, the total number of eggs and egg masses, percentage of eggs, and egg masses were highest on Baitiannuo and lowest on Tiancainuo. On the sweet maize, the total number of eggs and percentage of eggs on Jingtian No.3 were significantly higher than on Shuangsecuibao, Zaocuiwang, and Jingxiangyu (*p* < 0.05). On the waxy maize, the number of oviposited eggs was higher on Bainuo than on Heinuo and Ziyunuo. These results showed that the sweet–waxy maize was the preferred host for the female adults. Waxy maize, especially on Ziyunuo, was the least preferred.

#### 3.1.3. Oviposition Selectivity of *S. frugiperda* on Different Maize Varieties

The female adults laid the most eggs (710.00 ± 1.59 per cage), egg masses (5.50 ± 0.43 per cage) (Figure 3A), percentage of eggs (28.50%), and egg masses (26.44%) on Baitiannuo (Figure 3B), followed by Zaocuiwang, which had significant differences in the total number of eggs and percentage of eggs compared with other varieties (*p* < 0.05). This was followed by Wuke No.4 and Wuke 113, which were higher than those on Ziyunuo and Zhengdan 958. No significant difference was observed in the total number of egg masses and percentage of egg masses on Wuke No.4, Wuke 113, Ziyunuo, and Zhengdan 958 (*p* > 0.05). The lowest number of eggs (250.17 ± 1.62 per cage), egg masses (2.33 ± 0.21 per cage), percentage of eggs (10.04%), and egg masses (11.47%) occurred on Zhengdan 958. These results indicate that the *S. frugiperda* females significantly preferred to lay eggs on the special maize varieties over the common maize varieties, especially on Baitiannuo. Female adults had the lowest oviposition preference for Zhengdan 958.

### 3.2. Developmental Duration of Each Stage of S. frugiperda Feeding on Different Maize Varieties

The effect of the maize varieties on the developmental duration of *S. frugiperda* is presented in Table 1 and Table 2. *S. frugiperda* successfully completed its whole life history on all maize varieties, with the larval diet greatly affecting various developmental parameters. The developmental duration was significantly different among the six different varieties (Table 1). The development duration on the special maize varieties (Baitiannuo, Zaocuiwang, Ziyunuo) was shorter than that on the common maize varieties (Wuke No.4, Wuke 113, Zhengdan 958). The larval development period from the 1st to 6th instars was the longest on Zhengdan 958 and the shortest on Zaocuiwang, which was significantly shorter than on the common maize varieties (*p* < 0.05). The development duration of egg + larva (16.25 d), preadult (27.24 d), and total longevity (37.58 d) were shortest on Zaocuiwang and longest (21.31, 33.72, 43.46 days) on Zhengdan 958. The egg + larva, preadult, and total longevity when reared on different maize varieties had a similar order; from short to long, it was Zaocuiwang < Ziyunuo < Baitiannuo < Wuke No.4 < Wuke 113 < Zhengdan 958, respectively, which was significantly shorter the larvae fed on the special maize varieties than those on the common maize varieties (*p* < 0.05). Interestingly, no significant (*p* > 0.05) differences were found in the prepupa stage. The pupal stage varied significantly from 10.13 days (Zhengdan 958) to 8.77 days (Baitiannuo). The developmental duration of the adult stage was significantly shorter on Zhengdan 958 than that on others (*p* < 0.05) and there was no significant difference between the other maize varieties (*p* > 0.05).

The development duration of the larva, prepupa, and adult feeding on the six maize varieties varied according to gender (Table 2). When feeding on Wuke 113, the female 1st instar larvae development duration was significantly longer than that of the male (*p* < 0.05) and the male 3rd instar larvae was significantly longer than the female (*p* < 0.05). The female 2nd instar larvae development duration on Zaocuiwang was significantly longer than the male (*p* < 0.05). The female prepupa on Baitiannuo developed significantly faster than the males (*p* < 0.05). The female adults’ total longevity on all the six maize varieties was significantly longer than the male adults’ longevity (*p* < 0.05).

### 3.3. The Pupal Weight of S. frugiperda Reared on Different Maize Varieties

The *S. frugiperda* pupal weight reared on the various maize varieties is shown in Figure 4. The male pupal weight on the six maize hosts was higher than the female pupae. Significant differences occurred in the male pupal weight of *S. frugiperda* on the various maize varieties (*p* < 0.05). The male pupae weighed significantly more (*p* < 0.05) on Baitiannuo than on Wuke 113, Wuke No.4, and Zhengdan 958. The lowest pupal weight occurred on Zhengdan 958. The female pupae weight was greater on Baitiannuo. From the greatest to the least, the order of the pupal weight was Baitiannuo > Zaocuiwang > Ziyunuo > Wuke No. 4 > Wuke 113 > Zhengdan 958. The female and male pupae weight was significantly greater on the special maize varieties compared to the common maize varieties, except for the female pupal weight reared on Ziyunuo (*p* < 0.05).

### 3.4. Effects of Different Maize Varieties on Biological Parameters of Adults of S. frugiperda

The adult preoviposition period (APOP) and total preoviposition period (TPOP) of *S. frugiperda* on the special maize varieties were significantly shorter than on the common maize varieties (Table 3) (*p* < 0.05). The shortest APOP was 2.76 days on Baitiannuo and the longest was 4.37 days on Zhengdan 958. Similarly, the TPOP was shortest on Zaocuiwang (30.95 days) and longest on Zhengdan 958 (38.10 days). The longest oviposition days for the female moths was 4.43 days on Baitiannuo, which was significantly longer than on other maize varieties (*p* < 0.05). The shortest oviposition days occurred on Zhengdan 958 (3.00 days). The females’ mean fecundity on the special maize varieties was significantly higher than on the common maize varieties (*p* < 0.05). The mean fecundity varied significantly from 808.87 eggs on Baitiannuo to 456.44 eggs on Zhengdan 958. There was no significant difference in the female proportion (*p* > 0.05). The male proportion on Baitiannuo (0.37) was significantly (*p* < 0.05) higher than on Zhengdan 958 (0.26). The egg could easily hatch on the six maize varieties with a more than 90% hatching rate. Moreover, the hatching rate on the special maize varieties was higher than on the common maize varieties, and the highest hatching rate occurred on Baitiannuo (94.19%) and Zaocuiwang (94.55%), whereas the lowest was on Zhengdan 958 (92.02%).

### 3.5. Population Parameters of S. frugiperda Fed on Six Maize Varieties

The population parameters of *S. frugiperda* are shown in Table 4. The net reproductive rate (*R*_0_) on the special maize varieties was higher than on the common maize varieties. The finite rate of increase (*λ*) and intrinsic rate of increase (*r*) on the special maize varieties were significantly higher than on the common maize varieties (*p* < 0.05). The mean generation time (*T*) on the special maize varieties was significantly shorter than on the common maize varieties (*p* < 0.05). The highest *R*_0_ (355.01 offspring per female), *r* (0.1748 per day), and *λ* (1.1910 per day) occurred on Baitiannuo and the lowest *R*_0_ (157.22 offspring per female), *r* (0.1275 per day), and *λ* (1.1360 per day) were observed on Zhengdan 958. The *r* and *λ* were greater than 0 and 1, respectively, across all varieties. This is an indication that *S. frugiperda* can survive on these varieties. The *T* was 33.60–34.06 days on the special maize varieties and 35.39–39.66 days on the common maize varieties, among which the shortest *T* was on Zaocuiwang and the longest *T* on Zhengdan 958. The *T* of the *S. frugiperda* on the special maize varieties was significantly shortened (*p* < 0.05).

### 3.6. Age-Stage-Specific Survival Rate (S_xj_)

The age-stage-specific survival rate (*S_xj_*) on the six different maize varieties is shown in Figure 5. There were a lot of overlaps in the *S. frugiperda* stage-specific survivorship curves on the six maize varieties. The 1st and 2nd instar larvae survival rates were highest on Wuke 113 (100% and 91.11%, respectively) and lowest on Zaocuiwang (78.33% and 62.22%, respectively). The highest survival rates of the 3rd, 4th, and 6th instar larvae and prepupa were observed on Wuke 113 and the lowest on Wuke No.4. The 5th instar larvae survival rate was much higher on Baitiannuo than on the other maize varieties. The survival rates of the pupae, female adults, and male adults were highest on Baitiannuo (82.22%, 43.89%, and 36.67%, respectively), followed by Ziyunuo and Zaocuiwang. The lowest survival rates were recorded on Zhengdan 958 (62.78%, 34.44%, and 25%, respectively).

### 3.7. Population Survival Rate and Fecundity

The *l_x_* curve slowly decreased on the special maize varieties in the first 35 days, and slowly decreased from 100% to 78.33% (Baitiannuo), 72.78% (Ziyunuo), and 63.33% (Zaocuiwang), respectively. After 35 days, the *l_x_* rapidly decreased to 0. On the common maize varieties, the *l_x_* slightly decreased in the first 38 days and reduced from 100% to 64.44% (Wuke 113), 49.44% (Wuke No.4), and 60.00% (Zhengdan 958), respectively, whereas the *l_x_* quickly dropped to 0 within 9 days in the Wuke 113 group and 7 days on Wuke No.4 and 11 days on Zhengdan 958 (Figure 6).

The *f_x_*, *m_x_*, and *l_x_* × *m_x_* curves on all the maize varieties increased initially before decreasing. Reproduction began on Baitiannuo at 28 days, Ziyunuo at 28 days, Zaocuiwang at 27 days, Wuke 113 at 34 days, Wuke No.4 at 30 days, and Zhengdan 958 at 34 days. The *f_x_*, *m_x_*, and *l_x_* × *m_x_* curves reached maximum values on Baitiannuo (33 d), Ziyunuo (34 d), Zaocuiwang (32 d), Wuke 113 (38 d), Wuke No.4 (34 d), and Zhengdan 958 (40 d). The highest values occurred on Baitiannuo (155.0127, 84.4552, and 68.0333, respectively) and the lowest on Zhengdan 958 (92.7833, 57.9896, and 30.9278, respectively). The start of the reproduction age reached reproductive peaks on the special maize varieties earlier than on the common maize varieties (Figure 6).

### 3.8. Life Expectancy (e_xj_)

The age-stage-specific life expectancy is shown in Figure 7. The value of the *e_xj_* of all individuals feeding on all maize showed a downward trend and gradually decreased to 0 as age increased. *S. frugiperda* reared on Baitiannuo (34.33 d) had the longest *e_xj_* of newly laid eggs than on Wuke 113 (33.88 d), Ziyunuo (33.18 d), Zhengdan 958 (32.39 d), Zaocuiwang (31.71 d), and Wuke No. 4 (31.53). The highest *e_xj_* was observed on Zhengdan 958 in male adults (14.39 d) and female adults (15.24 d). The lowest *e_xj_* was observed on Baitiannuo in male (12.24 d) and Wuke 113 in female (13.70 d). The female adults’ *e_xj_* on the six maize varieties was higher than the male adults’ during the whole development stage.

### 3.9. Reproductive Value (v_xj_) of Each Age-Stage Group of Spodoptera frugiperda on Six Different Maize Varieties

The *S. frugiperda* reproductive value first increased and then decreased to 0 with the extension of the development stage (Figure 8). The curves of the *v_xj_* on each maize varieties significantly increased after adult emergence and had a peak; however, the pupal curve of the *v_xj_* on Ziyunuo, Wuke 113, and Zhengdan 958 had two peaks. The highest reproductive peak value of the female adults was 31 days on Baitiannuo (494.7667), 29 days on Zaocuiwang (374.7016), 37 days on Wuke 113 (368.7744), 31 days on Ziyunuo (362.7689), 32 days on WukeNo.4 (344.139), and 36 days on Zhengdan 958 (292.2861). The highest *v_xj_* occurred on Baitiannuo and the lowest *v_xj_* occurred on Zhengdan 958 (Figure 8).

## 4. Discussion

Insects lay eggs on highly suitable host plants to facilitate offspring development [32]. In the current study, *S. frugiperda* laid eggs on a number of special and common maize varieties. However, there were differences in oviposition preference. On the common maize varieties, the *S. frugiperda* oviposition preference was highest on Wuke No.4 and lowest on Zhengdan 958. On the special maize varieties, *S. frugiperda* females had a higher oviposition preference on Baitiannuo, and the lowest on Ziyunuo. Moreover, *S. frugiperda* female adults exhibited a significant oviposition preference on the special maize varieties compared to the common maize varieties. This suggests that *S. frugiperda* exhibited specific preferences in oviposition selectivity. Gripenberg et al. (2010) reported that offspring survived better on the plant type preferred by ovipositing females, as it improved their reproductive performance [32]. The higher *S. frugiperda* oviposition preference on Wuke No.4 and Baitiannuo could possibly be due to the presence of higher plant nutrients in these host plants, which is important for offspring development. Similar results showed that *S. frugiperda* female adults had a strong oviposition preference on maize compared with other test crops [33,34,35,36,37,38]. *S. frugiperda* females identify the most or least suitable maize landraces for the development of their offspring [14]. Compared with waxy and forage maize, *S. frugiperda* oviposition preference was higher on sweet maize [39]. Nascimento et al. (2020) reported that female moths showed a higher oviposition preference for transgenic plants over non-transgenic plants [40]. *S. frugiperda* showed a resilient oviposition preference for less-damaged Bt maize compared to damaged conventional maize [34]. Host preferences of insects seem to be influenced by plant characteristics that affect insect performance, including nutrients, allelochemicals, and even physical characteristics, such as hardness, size, shape, and texture [41]. Similar results indicated that diverse maize varieties can produce different herbivore-induced plant volatiles (HIPVs) profiles [42]. HIPVs are part of the indirect defense response of plants to herbivores. Herbivores could also find suitable ovipositing hosts by plant volatiles [41]. Yactayo-Chang et al. (2021) reported that *S. frugiperda* uses different clues of maize volatiles to determine the host location in its adult and larval stages [43]. Terpene volatiles induced by main herbivores may hinder the preference of *S. frugiperda* [42]. Nonanal and decanal, the volatiles of maize and rice, may be related to the oviposition selectivity of *S. frugiperda* [44]. *S. frugiperda* females, when laying eggs, will distinguish between different varieties, or the larvae will preferentially choose certain types of leaves [12,45]. This suggests that the oviposition preference on Zhengdan 958 and Ziyunuo may be due to the presence of physical or chemical characteristics, which may hinder the insect’s oviposition or fecundity.

Host plants play an important role in the growth, development, and reproduction of insects, and suitable hosts can improve the growth rate, survival rate, and fecundity of their offspring [46,47]. In the present study, *S. frugiperda* completed its life history on the various maize varieties. Zaocuiwang had the shortest egg + larval stage, preadult, and total longevity compared to others. The pupal stage, adult, and APOP were shortest and the fecundity and oviposition days were highest on Baitiannuo. The shortest TPOP and the highest hatching rate were observed on Baitiannuo and Zaocuiwang, respectively. The longest eggs + larvae, preadult, and adult stages, total longevity, APOP, and TPOP, and the lowest fecundity, oviposition days, and hatching rate occurred on Zhengdan 958. Significant differences in the mean developmental duration were observed between the adult and total longevity of the male and female adults when reared on the same maize varieties. These results indicated that the special maize varieties were more suitable for the growth and development of *S. frugiperda* than the common maize varieties. The populations were more adaptable on waxy maize than on the common maize varieties [48,49]. Conversely, Xu et al. (2021) reported the *S. frugiperda* larval stage and adult longevity on common maize varieties were shorter than on waxy maize [16]. This was attributed to the different maize breeding processes used in the experiments. The quality of the host plants as food for the insect larvae varies greatly among plant species [11]. Insects that feed on hosts with a low nutritional value can adopt compensation strategies by extending feeding periods or increasing food intake [50,51]. In the present study, *S. frugiperda* reared on the common maize varieties probably adapted their compensation strategies by extending the feeding period. Similar results had been reported that *S. frugiperda* larvae adapted their compensation strategy by prolonging their development duration [52,53]. The difference in the development time of *S. frugiperda* on different varieties may be attributed to the chemical characteristics of maize. Chemical substances in maize leaves affected the feeding selectivity of *S. frugiperda* [54]. Presently, a higher free fatty acid and lower water and amino acid content occurred on the special maize varieties. Similar studies have shown that the carbohydrate and protein content also play important roles in the development parameters of insects and may vary among the host plant [55,56]. In previous studies, low-quality protein and carbohydrates in larvae developed, but their efficiency in converting food into biomass was often low [57]. By comparing the consumption rate of *S. frugiperda* to different maize leaves, it was found that there might be differences in the composition of the host leaves [15].

The pupal state reflects the adaptability of larvae to a specific host or environment [58]. The pupal weight can reflect the insect’s appetite for the host plant. The pupal weight and fecundity of female adults of lepidoptera were positively correlated with their adaptability potential [45,59]. In this study, we found that the female adults’ pupal weight, pupal survival rate, and fecundity were higher on the special maize varieties than those on the common maize varieties. Compared to other maize varieties, the highest pupal weight, pupal survival rate, and fecundity were observed on Baitiannuo. The *S. frugiperda* larvae had the highest nutritional content on Baitiannuo. This may be due to variations in the physical and chemical characteristics of the host plant [60]. Previous studies have shown that differences in sebum composition in the upper and lower leaves of maize affected the feeding behavior of *S. frugiperda*. *S. frugiperda* larvae on leaves without epidermal lipids showed a higher weight and development time than on leaves containing epidermal lipids [61]. Trichomes on the leaf surface have been reported to have a resistant mechanism that inhibits the oviposition of female Chilo partellus [62]. Plant adaptability differences have also been reported to affect the *S. frugiperda* larva and pupal weight [15].

The biological parameter statistics, *R*_0_, *r*, *λ*, and *T*, integrate the growth, development, reproduction, and survival changes in insects, and indicate the ability of population growth in specific environments [63]. Significant differences occurred in the *S. frugiperda* population parameters on the various maize varieties. The order of value of the *R*_0_, *r*, and *λ* from high to low was Baitiannuo > Zaocuiwang > Ziyunuo > Wuke No.4 > Wuke113 > Zhengdan 958 (sweet–waxy maize > waxy maize > sweet maize > common maize). The *T* was shorter on the special maize varieties. The *r* and *λ* value were greater than 0 and 1, respectively. This indicates that various maize varieties supported the development of *S. frugiperda*. Comparatively, Baitiannuo was the most preferred host, whereas Zhengdan 958 was the least preferred host for *S. frugiperda*. The pests’ fitness to plants is attributed to the resistance ability of plants to insects [64]. Acharya et al. (2022) reported that the *S. frugiperda* net reproductive rate (*R*_0_) varied greatly among corn, rice, and potato [65]. Changes in the chemical and ultrastructural characteristics of plant surfaces may affect insect behavior, such as oviposition, orientation, and feeding [66]. Previous studies have shown that leaf toughness seemed to be a significant maize natural defense among different categories of germplasm [67]. Chuang et al. (2014) demonstrated that *S. frugiperda* saliva contains elicitors that trigger herbivore defenses in maize [68]. However, further studies are required to explore the mechanisms of different maize varieties against *S. frugiperda* growth and development.

To overcome the shortcomings of the traditional life table, the present study considered the age-stage, two-sex life table. The results show that the *S. frugiperda* survival rates were generally high on the six maize varieties. The stage survival curves had many overlaps and the varieties corresponding to the highest survival rates of each development stage were different. The survival rates of the larvae, except the 5th instar, were highest on Wuke 113, while the survival rates of the 5th instar larvae, pupae, female, and male adults were highest on Baitiannuo and lowest on Zhengdan 958. These results may be due to the difference in adaptability of the host plants at the larval stage [69]. Zhang et al. (2021) observed that the feeding selectivity of *S. frugiperda* larva to different maize was different in different development stages and feeding times [54]. The host plant chemical composition can be modified as a result of stress conditions [15,62], which can affect insect performance positively [63,64] or negatively [62,65]. Moreover, plants are known to contain secondary metabolites that are natural defense mechanisms against pathogenic invasion and herbivores and insect attacks [70]. This is an indication that the low performance of *S. frugiperda* on the common maize varieties, particularly Zhengdan 958, could be due to the presence of chemical or physical characteristics in the plants, which inhibited the performance of the *S. frugiperda*. Thus, local maize varieties, especially Zhengdan 958, could be used in areas where *S. frugiperda* is a major concern.

Sun et al. (2020) [49] found that *S. frugiperda* preferred Zhenghuangnuo No. 2 across all the other tested varieties, and that could be used to rear populations in the lab, plant in time as a lure belt, and spray insecticides to reduce the *S. frugiperda* population. In our study, *S. frugiperda* preferred the special maize varieties across all varieties, especially Baitiannuo. Thus, special maize varieties can be used to breed a large number of *S. frugiperda* populations in the laboratory, plant as a lure belt, and spray insecticides for the management of *S. frugiperda*. Many studies have speculated that when a preferred host is scarce and the *S. frugiperda* population density is high, *S. frugiperda* may transfer to other crops [64,71]. Therefore, it is necessary to closely monitor the occurrence of *S. frugiperda* to crops and determine the method of prevention.

## 5. Conclusions

This study showed that *S. frugiperda* successfully laid eggs and completed its life cycle across all maize varieties. However, *S. frugiperda* reared on the special maize varieties had higher oviposition preferences compared with the common maize varieties. Moreover, *S. frugiperda* had a shorter egg + larval stage, preadult, total longevity, pupal stage, adult, APOP, TPOP, and a higher pupal weight, fecundity, oviposition daily, and hatching rate on the special maize varieties. These led to greater *R*_0_, *r*, and *λ* and shorter *T* values on the special varieties. Although the common maize varieties were less preferable than the special maize varieties, *S. frugiperda* still completed its larva development and survival on them. Specifically, the low performance of *S. frugiperda* on the common maize varieties, particularly Zhengdan 958, could possibly be due to the presence of chemical or physical characteristics in the plants, which inhibited the performance of the *S. frugiperda*. We therefore recommend that further studies should be conducted to determine the presence of secondary metabolites in the resistant varieties. Moreover, common maize varieties, especially Zhengdan 958, could be used in areas where *S. frugiperda* is a major concern, but we should also pay close attention to the transfer hazards of *S. frugiperda* on other maize varieties. The findings of this study could be useful in predicting *S. frugiperda* population dynamics and provide suggestions for integrated pest management (IPM) strategies for *S. frugiperda* aimed at different maize varieties and help to refine laboratory rearing protocols.

## Figures and Tables

**Figure 1 insects-14-00413-f001:**
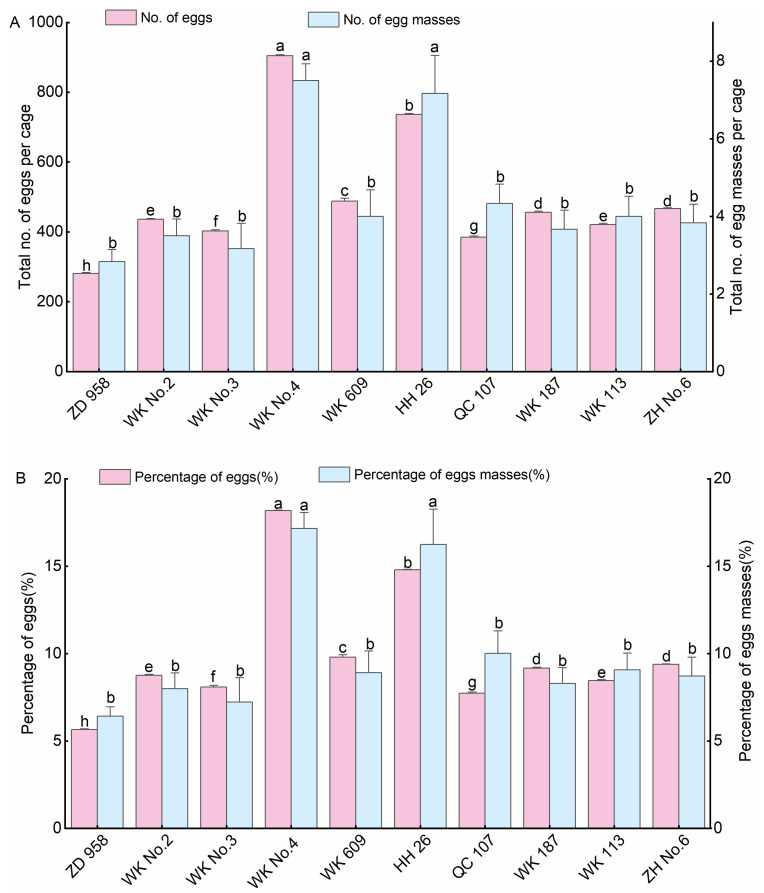
Oviposition selectivity of *Spodoptera frugiperda* on ten common maize varieties. Total number of eggs, egg masses per cage (**A**), and percentage of eggs and egg masses (**B**). ZD 958: Zhengdan 958, WK No.2: Wuke No.2, WK No.3: Wuke No.3, WK No.4: Wuke No.4, WK 609: Wuke 609, HH 26: Honghe 26, QC 107: Qingchu 107, WK 187: Wuke 187, WK 113: Wuke 113, ZH No.6: Zhenghong No.6, the same below. Data represent the mean ± SE of six replicates (*n* = 6). Different lowercase letters above the bars indicate significant differences by Duncan’s test (*p* < 0.05).

**Figure 2 insects-14-00413-f002:**
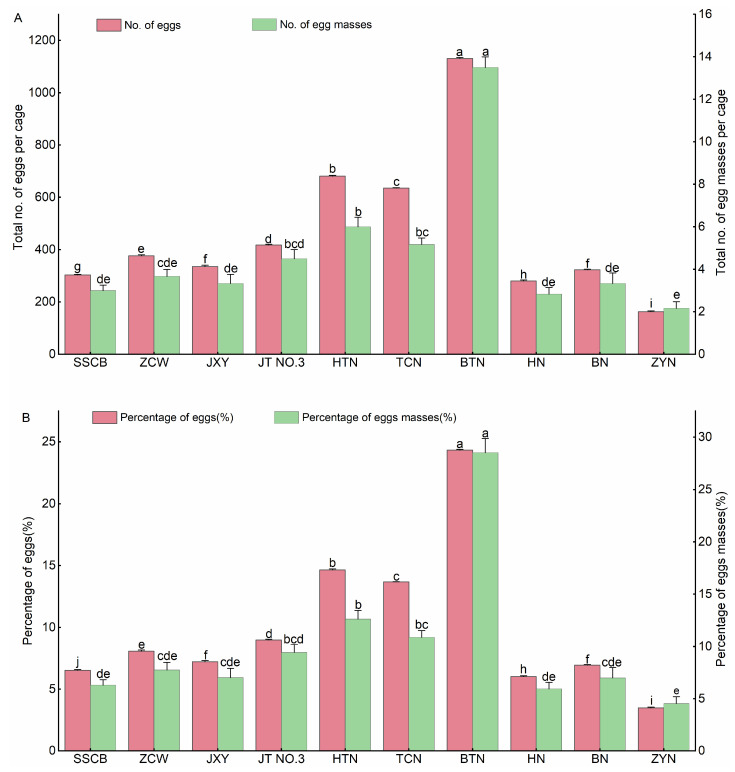
Oviposition selectivity of *Spodoptera frugiperda* on ten special maize varieties. (**A**,**B**) Mean ± standard error: (**A**) total number of eggs, egg masses per cage, and (**B**) average percentage of eggs and egg masses of *Spodoptera frugiperda* on ten special maize varieties from six experiments. SSCB: Shuangsecuibao, ZCW: Zaocuiwang, JXY: Jingxiangyu, JT NO.3: Jingtian No.3, HTN: Heitiannuo, TCN: Tiancainuo, BTN: Baitiannuo, HN: Heinuo, BN: Bainuo, ZYN: Ziyunuo, the same below. Different lowercase letters indicate significant differences among different special maize varieties by Duncan’s test (*p* < 0.05).

**Figure 3 insects-14-00413-f003:**
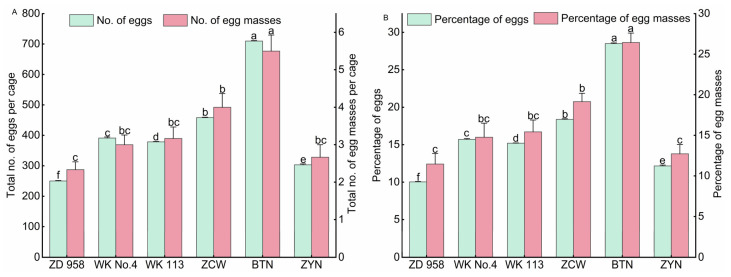
Oviposition selectivity of *Spodoptera frugiperda* on different types of maize varieties. Total number of eggs and eggs masses (**A**) and percentage of eggs and egg masses (**B**) laid by *Spodoptera frugiperda* on three common and three special maize varieties. Figure data are the mean ± SE of six replications. Significant differences are marked by different lowercase letters (*p* < 0.05).

**Figure 4 insects-14-00413-f004:**
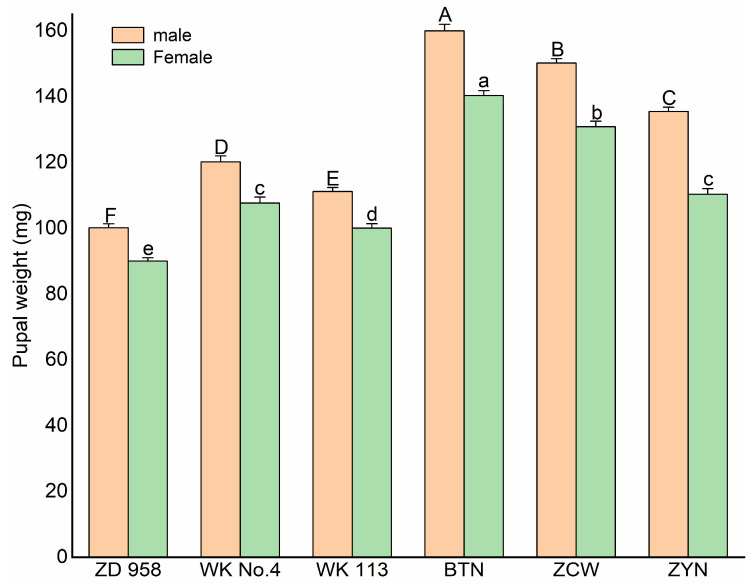
Male and female pupal weight of *Spodoptera frugiperda* feeding on six different maize varieties. The data are mean ± SE, different capital letters indicate significant differences for male pupal weight among different maize varieties by Duncan’s test (*p* < 0.05). Significant differences at the 0.05 level are marked by different lowercase letters for female pupal weight among different maize varieties (*p* < 0.05).

**Figure 5 insects-14-00413-f005:**
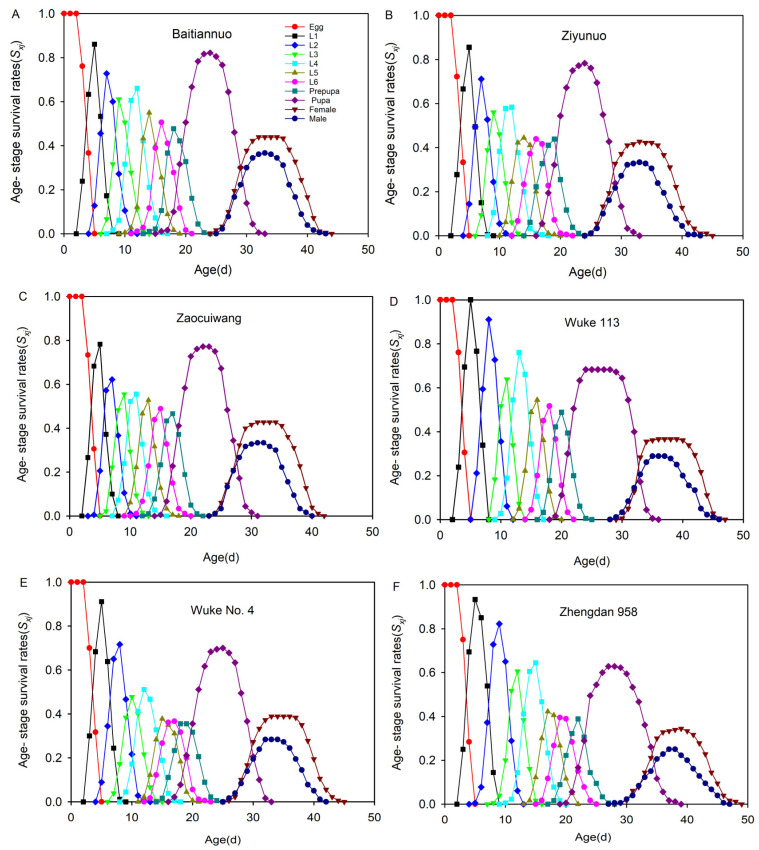
Age-stage-specific survival rate (*S_xj_*) of *Spodoptera frugiperda* reared on six different maize varieties. L1–L6 are the larval instars from first to sixth. The *S_xj_* on Baitiannuo (**A**), Ziyunuo (**B**), Zaocuiwang (**C**), Wuke 113 (**D**), Wuke No. 4 (**E**) and Zhengdan 958 (**F**).

**Figure 6 insects-14-00413-f006:**
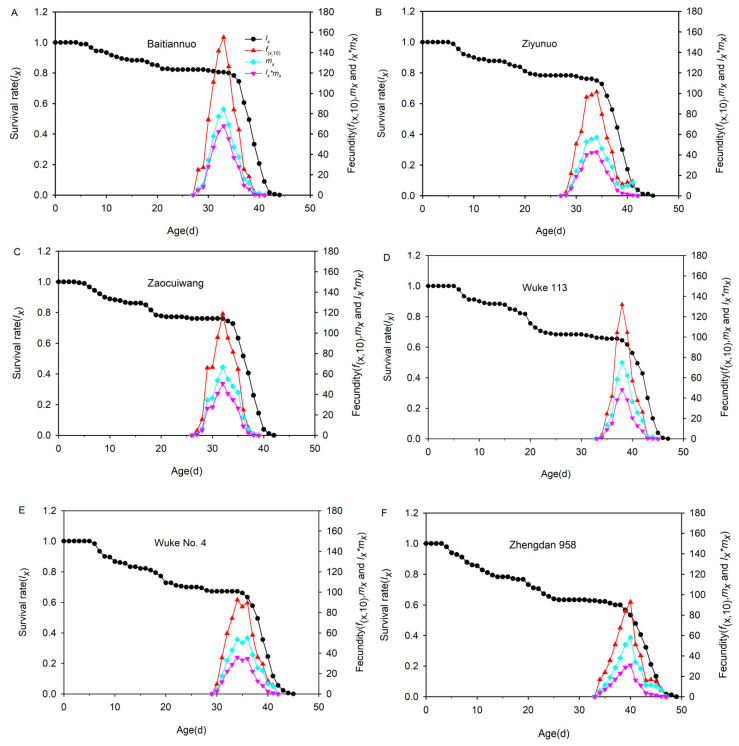
Age-specific survival rate (*l_x_*), female age-stage-specific fecundities (*f_x_*), age-specific fecundity (*m_x_*), and age-specific maternity (*l_x_* × *m_x_*) of *Spodoptera frugiperda* reared on six different maize varieties (Baitiannuo (**A**), Ziyunuo (**B**), Zaocuiwang (**C**), Wuke 113 (**D**), Wuke No. 4 (**E**) and Zhengdan 958 (**F**)).

**Figure 7 insects-14-00413-f007:**
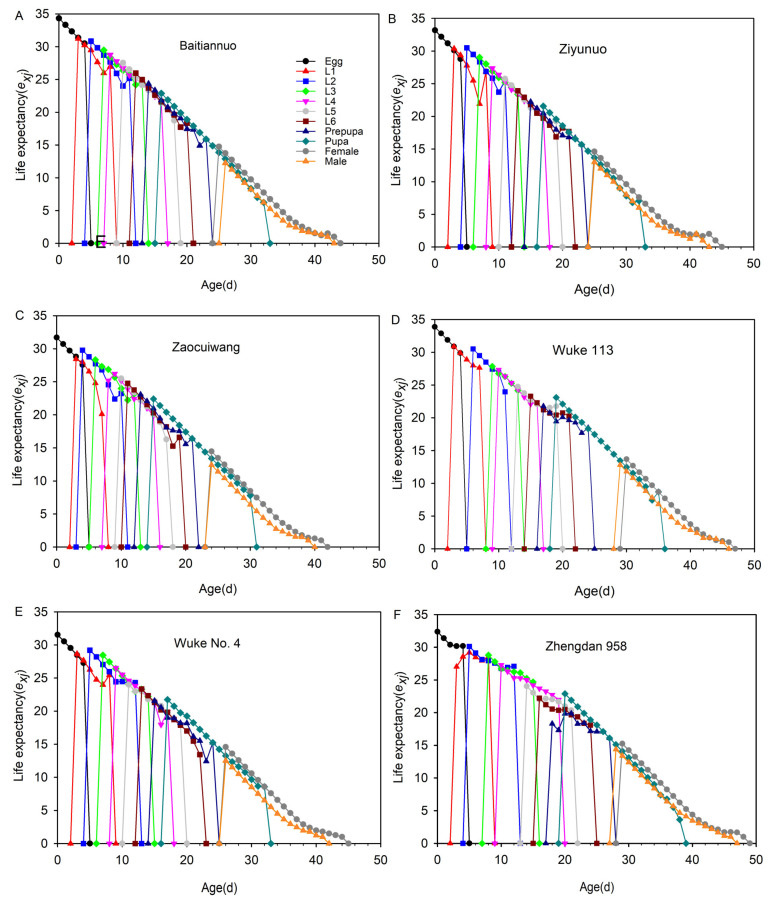
Age-stage-specific life expectancy (*e_xj_*) of *Spodoptera frugiperda* reared on six different maize varieties. L1–L6 represent 1st–6th instar larvae, respectively. The *e_xj_* on Baitiannuo (**A**), Ziyunuo (**B**), Zaocuiwang (**C**), Wuke 113 (**D**), Wuke No. 4 (**E**) and Zhengdan 958 (**F**).

**Figure 8 insects-14-00413-f008:**
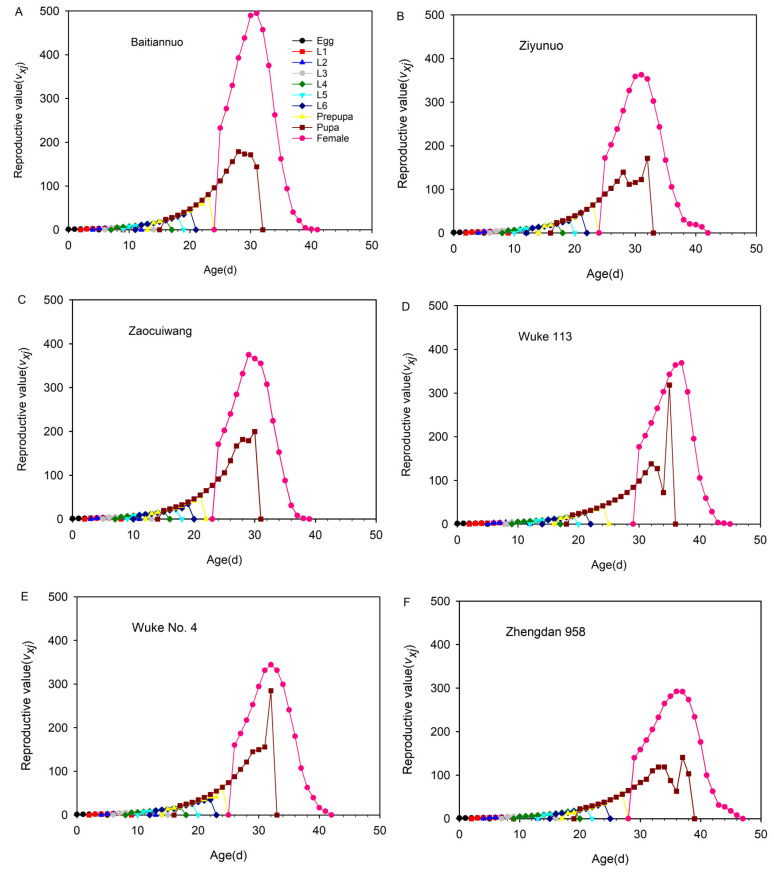
The age-stage reproductive value (*v_xj_*) of *Spodoptera frugiperda* on six different maize varieties. The *v_xj_* on Baitiannuo (**A**), Ziyunuo (**B**), Zaocuiwang (**C**), Wuke 113 (**D**), Wuke No. 4 (**E**) and Zhengdan 958 (**F**).

**Table 1 insects-14-00413-t001:** Developmental duration and longevity of different development stages of *Spodoptera frugiperda* reared on six maize varieties.

Stage (Days)	*n*	Baitiannuo	*n*	Zaocuiwang	*n*	Ziyunuo	*n*	Wuke No.4	*n*	Wuke 113	*n*	Zhengdan 958
Egg	180	4.13 ± 0.06 a	180	4.04 ± 0.56 a	180	4.06 ± 0.06 a	180	4.02 ± 0.06 a	180	4.07 ± 0.05 a	180	4.03 ± 0.05 a
1st instar	171	2.44 ± 0.04 d	167	2.40 ± 0.04 e	166	2.19 ± 0.03 d	163	2.77 ± 0.04 c	164	3.04 ± 0.15 b	160	3.61 ± 0.04 a
2nd instar	167	2.41 ± 0.04 d	161	2.01 ± 0.03 e	162	2.36 ± 0.04 d	155	2.77 ± 0.05 c	161	3.14 ± 0.03 b	149	3.45 ± 0.04 a
3rd instar	164	2.05 ± 0.03 b	159	1.89 ± 0.03 d	160	2.02 ± 0.03 bc	152	2.05 ± 0.03 b	159	1.96 ± 0.03 cd	144	2.37 ± 0.04 a
4th instar	161	2.62 ± 0.04 c	156	2.10 ± 0.03 d	158	2.56 ± 0.04 c	150	2.56 ± 0.04 c	155	2.98 ± 0.04 b	141	3.33 ± 0.04 a
5th instar	159	2.13 ± 0.04 ab	155	2.04 ± 0.04 b	157	2.10 ± 0.03 ab	147	2.05 ± 0.02 b	150	2.10 ± 0.03 ab	138	2.20 ± 0.04 a
6th instar	157	2.12 ± 0.03 b	154	2.04 ± 0.03 c	153	2.11 ± 0.03 bc	143	2.11 ± 0.03 bc	144	2.13 ± 0.03 b	134	2.24 ± 0.04 a
Egg + larva	157	17.87 ± 0.11 d	154	16.25 ± 0.11 e	153	17.59 ± 0.12 d	143	18.29 ± 0.13 c	144	19.42 ± 0.10 b	134	21.31 ± 0.13 a
Prepupa	148	2.25 ± 0.04 a	139	2.18 ± 0.04 a	142	2.23 ± 0.04 a	127	2.17 ± 0.03 a	125	2.26 ± 0.04 a	115	2.23 ± 0.04 a
Pupa	145	8.77 ± 0.05 d	137	8.82 ± 0.04 d	137	8.84 ± 0.04 d	121	9.08 ± 0.03 c	118	10.90 ± 0.06 a	109	10.13 ± 0.12 b
Preadult	145	28.81 ± 0.14 d	137	27.24 ± 0.13 e	137	28.58 ± 0.15 d	121	29.63 ± 0.15 c	118	32.63 ± 0.12 b	109	33.72 ± 0.20 a
Adult	145	10.26 ± 0.09 a	137	10.34 ± 0.10 a	137	10.33 ± 0.10 a	121	10.08 ± 0.10 a	118	10.24 ± 0.11 a	109	9.74 ± 0.12 b
Total longevity	145	39.06 ± 0.16 d	137	37.58 ± 0.17 e	137	38.91 ± 0.18 d	121	39.71 ± 0.18 c	118	42.86 ± 0.18 b	109	43.46 ± 0.23 a

Note: The data are mean ± SE, significant differences at the 0.05 level are marked by different lowercase letters in the same row.

**Table 2 insects-14-00413-t002:** Mean duration of each developmental stage of male and female of *Spodoptera frugiperda* on different maize varieties.

Stage (Days)	Sex	*n*	Baitiannuo	*n*	Zaocuiwang	*n*	Ziyunuo	*n*	Wuke No.4	*n*	Wuke 113	*n*	Zhengdan 958
Egg	♀	79	4.01 ± 0.09 Aab	77	4.00 ± 0.08 Aab	77	3.92 ± 0.09 Ab	70	3.99 ± 0.09 Aab	66	4.18 ± 0.09 Aa	62	3.98 ± 0.10 Aab
	♂	66	4.11 ± 0.09 Aab	60	3.93 ± 0.10 Ab	60	3.95 ± 0.10 Ab	51	4.00 ± 0.11 Aab	52	3.98 ± 0.10 Aab	47	4.21 ± 0.09 Aa
1st instar	♀	79	2.43 ± 0.06 Ad	77	2.21 ± 0.06 Ae	77	2.43 ± 0.06 Ad	70	2.86 ± 0.06 Ac	66	3.06 ± 0.03 Ab	62	3.68 ± 0.07 Aa
	♂	66	2.44 ± 0.07 Ad	60	2.18 ± 0.05 Ae	60	2.40 ± 0.07 Ad	51	2.75 ± 0.07 Ac	52	3.00 ± 0 Bb	47	3.70 ± 0.07 Aa
2nd instar	♀	79	2.43 ± 0.06 Ad	77	2.08 ± 0.04 Ae	77	2.31 ± 0.06 Ad	70	2.84 ± 0.07 Ac	66	3.11 ± 0.04 Ab	62	3.48 ± 0.07 Aa
	♂	66	2.38 ± 0.07 Ad	60	1.95 ± 0.04 Be	60	2.43 ± 0.07 Ad	51	2.67 ± 0.08 Ac	52	3.13 ± 0.05 Ab	47	3.34 ± 0.07 Aa
3rd instar	♀	79	2.00 ± 0.04 Ab	77	1.86 ± 0.05 Ac	77	2.00 ± 0.03 Ab	70	2.03 ± 0.05 Ab	66	1.86 ± 0.04 Bc	62	2.31 ± 0.06 Aa
	♂	66	2.06 ± 0.04 Ab	60	1.92 ± 0.04 Ac	60	2.08 ± 0.05 Ab	51	2.04 ± 0.05 Abc	52	2.06 ± 0.05 Ab	47	2.40 ± 0.08 Aa
4th instar	♀	79	2.70 ± 0.06 Ac	77	2.08 ± 0.04 Ad	77	2.53 ± 0.06 Ac	70	2.53 ± 0.07 Ac	66	3.02 ± 0.05 Ab	62	3.39 ± 0.07 Aa
	♂	66	2.52 ± 0.06 Ac	60	2.10 ± 0.06 Ad	60	2.63 ± 0.07 Ac	51	2.67 ± 0.08 Ac	52	2.98 ± 0.07 Ab	47	3.30 ± 0.07 Aa
5th instar	♀	79	2.09 ± 0.04 Ab	77	2.04 ± 0.06 Ab	77	2.12 ± 0.05 Ab	70	2.09 ± 0.03 Ab	66	2.15 ± 0.05 Aab	62	2.27 ± 0.06 Aa
	♂	66	2.18 ± 0.06 Aab	60	2.05 ± 0.05 Ab	60	2.05 ± 0.06 Ab	51	2.06 ± 0.03 Ab	52	2.12 ± 0.06 Aab	47	2.26 ± 0.06 Aa
6th instar	♀	79	2.14 ± 0.04 Aab	77	2.01 ± 0.05 Ac	77	2.10 ± 0.03 Ab	70	2.10 ± 0.04 Abc	66	2.12 ± 0.04 Aabc	62	2.24 ± 0.05 Aa
	♂	66	2.11 ± 0.04 Aa	60	2.07 ± 0.05 Aa	60	2.08 ± 0.04 Aa	51	2.14 ± 0.05 Aa	52	2.12 ± 0.04 Aa	47	2.15 ± 0.05 Aa
Egg + larva	♀	79	17.80 ± 0.16 Ad	77	16.27 ± 0.16 Ae	77	17.42 ± 0.17 Ad	70	18.43 ± 0.18 Ac	66	19.50 ± 0.12 Ab	62	21.35 ± 0.19 Aa
	♂	66	17.79 ± 0.15 Acd	60	16.20 ± 0.17 Ae	60	17.63 ± 0.18 Ad	51	18.31 ± 0.23 Ac	52	19.38 ± 0.20 Ab	47	21.36 ± 0.24 Aa
Prepupa	♀	79	2.18 ± 0.05 Bb	77	2.22 ± 0.05 Aab	77	2.23 ± 0.05 Aab	70	2.20 ± 0.05 Aab	66	2.35 ± 0.06 Aa	62	2.21 ± 0.05 Aab
	♂	66	2.33 ± 0.06 Aa	60	2.13 ± 0.06 Ab	60	2.22 ± 0.05 Aab	51	2.12 ± 0.05 Ab	52	2.19 ± 0.06 Aab	47	2.26 ± 0.07 Aab
Pupa	♀	79	8.73 ± 0.06 Ad	77	8.84 ± 0.05 Ad	77	8.86 ± 0.05 Ad	70	9.10 ± 0.05 Ac	66	10.92 ± 0.09 Aa	62	10.16 ± 0.15 Ab
	♂	66	8.80 ± 0.07 Ad	60	8.78 ± 0.07 Ad	60	8.82 ± 0.08 Ad	51	9.06 ± 0.05 Ac	52	10.87 ± 0.08 Aa	47	10.09 ± 0.18 Ab
Preadult	♀	79	28.71 ± 0.18 Ad	77	27.34 ± 0.18 Ae	77	28.51 ± 0.20 Ad	70	29.73 ± 0.19 Ac	66	32.77 ± 0.13 Ab	62	33.73 ± 0.25 Aa
	♂	66	28.92 ± 0.20 Ac	60	27.12 ± 0.19 Ae	60	28.67 ± 0.23 Acd	51	29.49 ± 0.24 Ac	52	32.44 ± 0.22 Ab	47	33.70 ± 0.32 Aa
Adult	♀	79	11.04 ± 0.08 Aab	77	11.14 ± 0.08 Aa	77	11.10 ± 0.09 Aa	70	10.86 ± 0.07 Ab	66	10.92 ± 0.09 Aab	62	10.52 ± 0.09 Ac
	♂	66	9.32 ± 0.08 Ba	60	9.30 ± 0.09 Ba	60	9.33 ± 0.12 Ba	51	9.02 ± 0.11 Bb	52	9.37 ± 0.14 Ba	47	8.72 ± 0.15 Bc
Total longevity	♀	79	39.75 ± 0.19 Ac	77	38.48 ± 0.19 Ad	77	39.61 ± 0.23 Ac	70	40.59 ± 0.20 Ab	66	43.70 ± 0.18 Aa	62	44.24 ± 0.25 Aa
	♂	66	38.24 ± 0.23 Bb	60	36.42 ± 0.22 Bc	60	38.00 ± 0.25 Bb	51	38.51 ± 0.25 Bb	52	41.81 ± 0.28 Ba	47	42.43 ± 0.37 Ba

Note: The data are mean ± SE, means followed by different lowercase letters in the same row are significantly different (*p* < 0.05), significant differences at the 0.05 level are marked by different capital letters in the same column for male and female at the same developmental stage.

**Table 3 insects-14-00413-t003:** Reproduction of *Spodoptera frugiperda* reared on six maize varieties.

Parameters	*n*	Baitiannuo	*n*	Zaocuiwang	*n*	Ziyunuo	*n*	Wuke No.4	*n*	Wuke 113	*n*	Zhengdan 958
APOP(d)	79	2.76 ± 0.07 d	77	3.61 ± 0.06 c	77	3.53 ± 0.06 c	70	3.99 ± 0.08 b	66	4.35 ± 0.07 a	62	4.37 ± 0.08 a
TPOP(d)	79	31.47 ± 0.19 e	77	30.95 ± 0.19 e	77	32.04 ± 0.21 d	70	33.72 ± 0.23 c	66	37.12 ± 0.15 b	62	38.10 ± 0.26 a
Oviposition days (d)	79	4.43 ± 0.07 a	77	3.83 ± 0.05 bc	77	3.92 ± 0.07 b	70	3.71 ± 0.07 c	66	3.38 ± 0.06 d	62	3.00 ± 0.06 e
Fecundity (eggs/female)	79	808.87 ± 6.54 a	77	621.03 ± 6.40 b	77	609.64 ± 8.48 b	70	555.14 ± 12.95 c	66	523.50 ± 12.01 c	62	456.44 ± 7.80 d
Female proportion (Nf/N)	180	0.44 ± 0.04 a	180	0.43 ± 0.04 a	180	0.43 ± 0.04 a	180	0.39 ± 0.04 a	180	0.37 ± 0.04 a	180	0.34 ± 0.04 a
Male proportion (Nm/N)	180	0.37 ± 0.04 a	180	0.33 ± 0.04 ab	180	0.33 ± 0.04 ab	180	0.28 ± 0.04 ab	180	0.29 ± 0.03 ab	180	0.26 ± 0.03 b
Hatching rate	79	0.9419 ± 0.0018 a	77	0.9455 ± 0.0011 a	77	0.9339 ± 0.0016 b	70	0.9228 ± 0.0014 c	66	0.9327 ± 0.0016 b	62	0.9202 ± 0.0011 c

Note: Values (mean ± SE) followed by different lowercase letters in the same rows indicate a significant difference (*p* < 0.05).

**Table 4 insects-14-00413-t004:** Population parameters of *Spodoptera frugiperda* reared on six different maize varieties.

Hosts	Net Reproductive Rate (*R*_0_) (Offspring/Individual)	Intrinsic Rate of Increase (*r*) (d^−1^)	Finite Rate of Increase (*λ*) (d^−1^)	Mean Generation Time (*T*) (d)
Baitiannuo	355.01 ± 30.04 a	0.1748 ± 0.0027 a	1.1910 ± 0.0033 a	33.60 ± 0.19 c
Zaocuiwang	265.66 ± 22.98 b	0.1702 ± 0.0029 ab	1.1856 ± 0.0034 ab	32.79 ± 0.20 d
Ziyunuo	260.79 ± 22.75 b	0.1634 ± 0.0028 b	1.1775 ± 0.0033 b	34.06 ± 0.21 c
Wuke No.4	215.89 ± 20.79 bc	0.1519 ± 0.0030 c	1.1638 ± 0.0035 c	35.39 ± 0.24 b
Wuke 113	191.95 ± 19.27 cd	0.1345 ± 0.0027 d	1.1440 ± 0.0030 d	39.08 ± 0.16 a
Zhengdan 958	157.22 ± 16.38 d	0.1275 ± 0.0028 d	1.1360 ± 0.0032 d	39.66 ± 0.27 a

Note: Data are mean ± SE. Significant differences are expressed by different lowercase letters in the same column (*p* < 0.05).

## Data Availability

The datasets in this study are available from the corresponding author on reasonable request.

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
