# Peer review of "Oviposition Preference and Age-Stage, Two-Sex Life Table Analysis of Spodoptera frugiperda (Lepidoptera: Noctuidae) on Different Maize Varieties"

_insects, 2023, doi:10.3390/insects14050413_

Round 1

Reviewer 1 Report

Title: remove "reared"
Conclusions on summary, abstract and main texts are not much compatible. 

Introduction: 63-65: having some facts (data) could support this statement better. Now, the statement is too general. I was expected to see some production, losses, etc. data there.  

Discussion:
In summary and abstract authors said, this study gives a simple basic scientific information/rear protocol, but in discussion (L522-523), they are recommending a maize variety of the farmers. That was not well explained in introduction, summary and abstract, and in discussion. 

Whole MS: While I scanned this MS by TurnItIn, I got more than 20% similarity. I suggest authors to take care of those. 

Author Response

We have carefully revised our manuscript according to your valuable comments (please see the attachment and revised manuscript)

Reviewer 2 Report

This is an interesting article on the life cycle of an important pest of corn crops. Well conducted methodically with interesting results.

A paragraph on the practical use of the results of this research should be added to the discussion (there is currently one sentence in the conclusions).

I also have a question about the temperature used in the experiment. Typically,  different temperatures/temperature ranges are used in these types of experiments. It is known that temperature has a decisive influence on the growth and yield of plants. Why was the temperature of 25 ± 1 °C chosen for the study? Please, explain in the material and methods section.

Other notes:

line 47 - should be Poaceae, Asteraceae and Fabaceae

line 53 - This sentence shows that China is an African country.

Author Response

We have carefully revised our manuscript according to your valuable comments (please see response letter and revised manuscript)

Round 2

Reviewer 1 Report

Authors upgraded the MS became better.

Author Response

Thanks for your positive comments and constructive suggestions!